# Dietary Calcium Intake and Hypertension: Importance of Serum Concentrations of 25-Hydroxyvitamin D

**DOI:** 10.3390/nu11040911

**Published:** 2019-04-23

**Authors:** Haruki Nakamura, Hiromasa Tsujiguchi, Akinori Hara, Yasuhiro Kambayashi, Sakae Miyagi, Thao Thi Thu Nguyen, Keita Suzuki, Yuichi Tao, Yuriko Sakamoto, Yukari Shimizu, Norio Yamamoto, Hiroyuki Nakamura

**Affiliations:** 1Department of Environmental and Preventive Medicine, Graduate School of Medical Science, Kanazawa University, Ishikawa 920-8640, Japan; t-hiromasa@med.kanazawa-u.ac.jp (H.T.); ahara@m-kanazawa.jp (A.H.) ykamba@med.kanazawa-u.ac.jp (Y.K.); smiyagi@staff.kanazawa-u.ac.jp (S.M.); toi_fs@yahoo.com (T.T.T.N.); keitasuzuk@yahoo.co.jp (K.S.); yuichi.tao@stu.kanazawa-u.ac.jp (Y.T.); birthplace3119@i.softbank.jp (Y.S.); h_zu@me.com (Y.S.); hiro-n@po.incl.ne.jp (H.N.); 2Department of Orthopaedic Surgery, Graduate School of Medical Science, Kanazawa University, Ishikawa 920-8640, Japan; norinori@med.kanazawa-u.ac.jp

**Keywords:** hypertension, calcium, vitamin D, 25-hydroxyvitamin D

## Abstract

The relationship among dietary calcium, hypertension and vitamin D status currently remains unclear. This population-based cross-sectional study examined the association between dietary calcium intake and hypertension and the influence of serum concentrations of 25-hydroxyvitamin D [25(OH)D] in Japanese subjects. A total of 619 subjects aged from 40 years were recruited. Dietary intake was measured using a validated brief self-administered diet history questionnaire. Hypertension was defined as the use of antihypertensive medication or a blood pressure of 140/90 mmHg. Serum concentrations of 25(OH)D were used as the biomarker of vitamin D status. The prevalence of hypertension and low serum 25(OH)D levels (<20 ng/mL) were 55 and 32%, respectively. Dietary calcium intake inversely correlated with hypertension in subjects with serum 25(OH)D levels higher than 20 ng/mL (OR: 0.995; 95% CI: 0.991, 0.999) but it was not significant in those with serum 25(OH)D levels of 20 ng/mL or lower. Furthermore, dietary vitamin D intake correlated with serum concentrations of 25(OH)D after adjustments for various confounding factors. The present results demonstrate that the regular consumption of calcium may contribute to the prevention and treatment of hypertension in subjects with a non-vitamin D deficiency and also that dietary vitamin D intake may effectively prevents this deficiency.

## 1. Introduction

Hypertension is one of the main risk factors for cardiovascular morbidity and mortality worldwide [1]. Lifestyle modifications are a key strategy for the prevention and treatment of hypertension. Previous studies have reported that several lifestyle habits, such as reduced sodium intake [2], weight loss [3], moderation in alcohol consumption [4] and increased physical activity [5], contribute to reducing blood pressure (BP).

Increases in calcium intake have been shown to lower BP in hypertensive and normotensive subjects. Dietary calcium interventions, such as supplementation or food fortification, were found to significantly reduce systolic blood pressure (SBP) and diastolic blood pressure (DBP) in normotensive subjects [6]. A meta-analysis of prospective cohort studies revealed that higher calcium intake from food and supplements is associated with lower risk of hypertension [7]. Calcium intake has been considered to have the effect of inhibiting the renin-angiotensin system and reducing the contractions of vascular smooth muscles cells [8,9]. Although the hypotensive effects of dietary calcium intake are weak, small reductions in BP are considered to have important health implications for cardiovascular diseases.

A close relationship exists between calcium and vitamin D. A sufficient vitamin D status promotes the intestinal calcium absorption and inhibits its excretion from the kidneys [10]. For example, osteoporosis, which is characterized by low bone mass and deterioration of bone tissue, is mainly associated with an inadequate calcium intake. However, an insufficient vitamin D status also contributes to osteoporosis due to the associated reductions in calcium absorption [11]. The relationship between vitamin D intake and BP has been examined but currently remains unclear [12,13].

To examine the association between dietary calcium and hypertension, it is important to consider the role of vitamin D. However, few studies have investigated this triangular relationship. A previous study reported that dietary calcium and dietary vitamin D were related to reductions in the risk of hypertension [14]. Dietary vitamin D intake and vitamin D from cutaneous synthesis reflect on serum concentrations of 25-hydroxyvitamin D [25(OH)D], which is the biomarker of vitamin D status [10]. Although serum 25(OH)D concentrations are inversely associated with hypertension [15,16], few population studies have been conducted on the association among dietary calcium intake, serum concentrations of 25(OH)D and hypertension.

Therefore, the present cross-sectional study aimed to examine the association among dietary calcium intake, hypertension and serum concentrations of 25(OH)D.

## 2. Methods

### 2.1. Study Design and Participants

Comprehensive health examination data gathered between March 2014 and January 2016 from the residents of Shika town, Ishikawa Prefecture, a rural area in Japan, were used in the present cross-sectional study. These baseline data were derived from the SHIKA study, which is a longitudinal population-based observational study conducted to describe the health status of the population in a town model and investigate approaches to prevent life-related diseases, such as hypertension, diabetes, allergic diseases and liver diseases. This study population has been reported previously [17]. In brief, there were 22,314 residents in Shika town in March 2014, 18,839 of whom were middle-aged (40 years or older) [18]. The target subjects of the SHIKA study were all middle-aged residents who were legally residing in two specific elementary school districts and there were 2160 residents in these districts in March 2014. All subjects in the model districts (*n* = 2160) were delivered a self-administrated questionnaire by researchers, who explained the outline of the SHIKA study and the way to complete the questionnaire. The response rate of the questionnaire was approximately 92% (*n* = 1987). We recruited 837 voluntary subjects who underwent the comprehensive health examination. 

The data of 787 subjects were available for the present study. To avoid under/overestimations leading to bias in the analysis of other nutrients, all subjects who reported a total energy intake of less than 600 kcal/day (half of the energy intake required for the lowest physical activity category) or more than 4000 kcal/day (1.5-fold the energy intake required for the moderate physical activity category) were excluded (*n* = 9). In addition, to weaken the influence of a treatment bias, subjects with diabetes, dyslipidaemia, cerebrovascular diseases and cardiovascular diseases were excluded (*n* = 159). Informed consent was obtained from all subjects. The present study was ultimately conducted on 619 subjects.

### 2.2. Nutrition Status

Nutrition status was assessed using a brief-type self-administered diet history questionnaire (BDHQ). BDHQ was a brief version of the diet history questionnaire and asked subjects about the consumption frequency of 58 food and beverage items. These items are commonly consumed in Japan and mainly from the food list used in the National Health and Nutrition Survey of Japan. BDHQ was developed in Japan for use in large-scale nutritional epidemiology studies. A previous study compared estimated energy and nutrient intakes calculated using BDHQ with 16-day dietary records in a Japanese population; the energy-adjusted intakes of 42 nutrients correlated with dietary records and Pearson’s correlation coefficients ranged between 0.45 and 0.61 in women and 0.41 and 0.63 in men [19]. Other studies have also confirmed the validity of BDHQ, which is considered to have a satisfactory ranking ability for many nutrients in Japanese subjects [20,21].

### 2.3. Serum 25(OH)D Measurement

Serum concentrations of 25(OH)D were used as the biomarker of vitamin D status. Vitamin D obtained from food and sun exposure is biologically inert and undergoes hydroxylation twice for activation. Vitamin D is converted to 25(OH)D in the liver and 25(OH)D is then metabolized to the biologically active form of 1,25-dihidroxyvitamin D in the kidneys. Serum concentrations of 25(OH)D reflect vitamin D obtained from food and supplements as well as that synthesized in the skin and has a long circulating half-life of 15 days, while 1,25-dihidroxyvitamin D has a short half-life of 15 hours and is strictly regulated by calcium, phosphate and the parathyroid hormone [22].

Serum concentrations of 25(OH)D were measured using a radioimmunoassay (25-Hydroxyvitamin D 125I RIA Kit, DiaSorin Inc., Stillwater MN, USA). This method is based on an antibody with specificity to 25(OH)D and is considered to be extremely sensitive and specific [21]. According to the Institute of Medicine, serum concentration of 25(OH)D ≥20 ng/mL (50 nmol/L) is sufficient for maintaining bone health [10]. In the present study, low serum 25(OH)D concentration was defined as a 25(OH)D value of 20 ng/mL (50 nmol/L) or lower.

### 2.4. BP Measurement

BP was measured in the comprehensive health examination for all subjects by well-trained nurses and clinical technologists, who were the staff of specialized medical check-up centres (ISHIKAWA HEALTH SERVICE ASSOCIATION, Ishikawa, Japan) and completed training courses on the BP measurement protocol. Subjects sat in a chair at rest and BP was measured twice consecutively using the right upper arm with a suitably sized cuff attached to UM-15P (Parama-tech Co., Ltd., Fukuoka, Japan) and HEM-907 (OMRON Co., Ltd., Kyoto, Japan), an automated digital sphygmomanometer based on the oscillometric method. Averages were adopted as BP data in the present study and hypertension was defined as a BP of 140/90 mmHg or higher. Subjects being treated with antihypertensive drugs were also included as hypertensive subjects.

### 2.5. Other Variables

Weight, height and waist circumference were measured and blood samples were taken in the comprehensive health examination for all subjects. Body mass index (BMI) was calculated as weight in kilograms divided by the square of height in meters.

Self-administered questionnaires were used to assess daily habits. The frequency of exercise was classified into two groups according to answers to the following questions: “Do you have a habit of exercise for more than 30 minutes at least two times a week for 1 year?” or “Do you habitually perform tasks such carrying baggage, walking and cleaning for more than 1 hour a day daily?” An exercise habit was defined by replying in the affirmative to either of these questions in the present study. Smoking habit was classified into two groups based on whether subjects were current smokers or not.

### 2.6. Statistical Analysis

The Student’s *t*-test was used to compare the average of continuous variables and the chi-square test was used to compare the proportions of categorical variables. All subjects were stratified into two groups: BP groups (Hypertension group and Normal BP group) and 25(OH)D groups (normal serum 25(OH)D level and low serum 25(OH)D level). A two-way analysis of variance (two-way ANOVA) was used to examine differences in dietary calcium intake between the BP groups and 25(OH)D groups. A multiple logistic regression analysis after adjustments for the following independent factors: sex, age, BMI, estimated glomerular filtration rate (eGFR), smoking status, exercise habit, energy intake, serum concentration of 25(OH)D, sodium intake, consumption of alcohol, protein, carbohydrates, total dietary fibre, saturated fatty acids, monounsaturated fatty acids, n-3 fatty acids and n-6 fatty acids, was performed to assess the association between BP and dietary calcium intake. Furthermore, to investigate the association between dietary vitamin D intake and serum concentrations of 25(OH)D, we used a multiple linear regression analysis controlling for sex, age, BMI, exercise habit, alcohol intake, smoking status, eGFR. In the multiple linear regression analysis, all data were transformed to z scores to obtain standardized regression coefficients.

Statistical significance was considered with an α critical value of 0.05. Statistical analyses were performed using IBM SPSS Statistics version 24.0 for Mac (SPSS Inc., Armonk, NY, USA).

### 2.7. Ethics Statement

This study was conducted with the approval of the Ethics Committee of Kanazawa University. Written informed consent was obtained from all participants.

## 3. Results

The basic characteristics of subjects according to BP groups were shown in Table 1. In 619 subjects with a mean age of 60.9 years, the mean SBP and DBP and intake of calcium were 139 mmHg and 80.7 mmHg and 292 mg/1000 kcal/day, respectively. Among 343 hypertensive subjects, 182 were being treated with antihypertensive drugs. Hypertensive subjects were significantly older, showed a male predominance, had a higher BMI and had lower eGFR and serum concentration of 25(OH)D. No significant differences were observed in energy intake levels. Among individual nutrients, hypertensive subjects consumed more sodium, alcohol, saturated fatty acids and monounsaturated fatty acids and less n-6 fatty acids than normotensive subjects. No significant differences were noted in the dietary intake of calcium and vitamin D.

The two-way ANOVA was used to stratify the association between dietary calcium in the BP groups according to 25(OH)D groups (Table 2). The prevalence of low serum 25(OH)D level was 32% (*n* = 201). The obtained results revealed a significant interaction between BP groups and 25(OH)D groups for dietary calcium intake (*p* = 0.015). In subjects with a low serum 25(OH)D level, the consumption of dietary calcium was higher in normotensive than in hypertensive subjects. On the other hand, in subjects with a normal serum 25(OH)D level, the consumption of dietary calcium was higher in hypertensive than in normotensive subjects. However, these differences were not significant in the Student’s t-test; *p* values were 0.067 for a low serum 25(OH)D level and 0.099 for a normal serum 25(OH)D level.

To adjust for the effects of confounding factors, a multiple logistic regression analysis was used to evaluate the association between dietary calcium intake and blood pressure groups. Based on the significant interaction between BP groups and 25(OH)D groups for dietary calcium intake (Table 2), we performed separate multiple logistic regression analyses according to serum 25(OH)D levels. The results obtained are shown in Table 3. In subjects with a normal 25(OH)D level, dietary calcium intake was inversely associated with hypertension after adjustments for the following confounding factors: sex, age, BMI, eGFR, exercise habit, smoking status, serum 25(OH)D, sodium intake and consumption of alcohol, protein, carbohydrates, total dietary fibre, saturated fatty acids, monounsaturated fatty acids, *n*-3 fatty acids and *n*-6 fatty acids. In subjects with a low 25(OH)D level, there was no significant association between dietary calcium intake and hypertension. These associations were the same in other multiple regression analysis models.

Figure 1 shows the spread of the association between dietary vitamin D and serum 25(OH)D concentration. A correlation was observed between dietary vitamin D and serum 25(OH)D concentration and Pearson’s correlation coefficient was 0.285 (*p* <0.001). This association was also examined using a multiple linear regression analysis; the association between dietary vitamin D intake and serum 25(OH)D concentration was significant after adjustments for the following confounding factors: age, sex, exercise habit, smoking status, eGFR and the consumption of alcohol (Table 4).

## 4. Discussion

The present cross-sectional study was conducted in an attempt to examine the association between dietary calcium intake and hypertension depending on the presence or absence of a vitamin D deficiency. Serum concentrations of 25(OH)D were used as the biomarker of vitamin D status. The results obtained suggested that high dietary calcium intake is inversely associated with hypertension in subjects with normal serum 25(OH)D level. Furthermore, a correlation was observed between dietary vitamin D intake and serum 25(OH)D levels after adjusting for various confounding factors.

Our result is consistent with previous findings. A meta-analysis that selected trials on randomized normotensive individuals receiving dietary calcium food fortification and supplementation versus a placebo or control showed that an increased calcium intake significantly reduced SBP and DBP and the extent of this hypotensive effect was -1.43 mmHg and -0.98 mmHg, respectively [6]. Another meta-analysis that selected prospective observational studies investigating the relationship between calcium intake and the risk of developing hypertension reported that a higher calcium intake correlated with a lower risk of developing hypertension in the general population [7]. Although the odds ratio of dietary calcium intake and hypertension in the present study was small, in previous studies the reductions observed in BP by an increased calcium intake were also small and small reductions in BP and in the risk of hypertension have important health implications for cardiovascular diseases. For example, previous studies based on the Framingham Heart Study and National Health and Nutrition Examination Survey II reported that only a 2 mmHg reduction in DBP resulted in a 17% decrease in the prevalence of hypertension, a 6% reduction in coronary heart disease and a 15% reduction in stroke and transient ischemic attack in the general population [23]. A small reduction in the distribution of SBP is also considered to result in a marked reduction in mortality due to hypertension-related diseases, such as coronary heart disease and stroke [24]. A population-based approach is an important strategy and the present results support the effectiveness of dietary calcium intake for the prevention and treatment of hypertension, particularly in subjects with a non-vitamin D deficiency.

Since vitamin D is crucially involved in the absorption of dietary calcium [25], we examined the association between dietary calcium intake and hypertension in consideration of the effects of serum concentrations of 25(OH)D. In the present study, a significant interaction was demonstrated between hypertension and serum 25(OH)D levels for dietary calcium intake. To the best of our knowledge, few population studies have investigated the association among dietary calcium intake, serum concentrations of 25(OH)D and hypertension. The joint association of dietary calcium intake and dietary vitamin D intake with the risk of hypertension was investigated in the previous study but was not significant. [12]. However, a vitamin D deficiency is not evaluated by the intake of vitamin D but rather serum 25(OH)D. Furthermore, 1,25-dihydroxyvitamin D, which is the active form of vitamin D and is metabolized from 25(OH)D in the kidneys, is not suitable for the diagnosis of a vitamin D deficiency because its serum level may be elevated by secondary hyperparathyroidism [25]. We consider the absence of similar findings to not necessary deny the contribution of a vitamin D deficiency assessed by serum concentrations of 25(OH)D to the inverse association between dietary calcium intake and hypertension.

The mechanisms responsible for the impact of calcium intake on BP have not yet been elucidated in detail. Previous findings showed that dietary calcium reduced the contractions of vascular smooth muscle cells [8,26] and inhibited the renin-angiotensin system [9,27]. Furthermore, improvements in the sodium-potassium balance [28] and enhanced insulin sensitivity [29] were suggested to contribute to a reduction in BP. In the present study, an inverse association between dietary calcium and hypertension was only observed in subjects with normal serum 25(OH)D level. This is biologically explainable because calcium absorption requires vitamin D; without vitamin D, only 10% of dietary calcium is absorbed [25]. Furthermore, a previous study reported that serum 25(OH)D levels correlated with intestinal calcium transport; when serum concentrations of 25(OH)D increased from 20 to 32 ng/mL, intestinal calcium transport increased by 45–65% [30]. We consider dietary calcium intake to potentially exert hypotensive effects; however, its benefit is markedly attenuated by a vitamin D deficiency.

Vitamin D deficiency and insufficiency is common in many populations worldwide. For example, the prevalence of a 25(OH)D value of 20 ng/mL (50 nmol/L) or lower was previously reported to be approximately 41% in outpatients aged between 49 and 83 years and 36% in healthy young adults aged between 18 and 29 years in the United States [31]. The prevalence of the risk of vitamin D deficiency is also high in Japan. A previous population-based cross-sectional study showed that the proportions of 9084 participants with a serum concentration of 25(OH)D <20 ng/mL (50 nmol/L) and ≥30 ng/mL (75 nmol/L) were 53.6 and 9.1%, respectively [32]. In the present study, even after taking into consideration the selection bias that subjects were voluntary collaborators of the comprehensive health examination and, thus, were more likely to be health conscious, the prevalence of a 25(OH)D value of 20 ng/mL (50 nmol/L) or lower was as high as in previous studies (201 out of 619 subjects, 32%). 

The primary source of vitamin D is solar ultraviolet B radiation (wavelength of 290–315 nm) [33]. Ultraviolet B radiation penetrates the skin and converts 7-hydrocholesterol in the skin to pre-vitamin D3, which is transformed into vitamin D3. The research field of the present study, Ishikawa prefecture, has less daylight hours than other prefectures in Japan [34] and complete cover and shade has been reported to reduce the energy of ultraviolet B radiation by 50% [35]. Humans also obtain vitamin D from their diet; dietary sources of vitamin D are natural foods, such as salmon, mackerel and egg yolk, as well as fortified food and supplements [36]. In the present study, a correlation was observed between dietary vitamin D intake and serum concentrations of 25(OH)D. This result supports the importance of a high dietary vitamin D intake in addition to exposure to sunlight for preventing vitamin D deficiency.

Several limitations of the present study need to be acknowledged. First, we were unable to assess causality because of the cross-sectional nature of the present study and the size of the study subjects was not large enough to elucidate the relationship. Further studies are required to confirm our result. Second, selection bias needs to be considered; the ratio of health-conscious individuals may have been high among subjects because they were voluntary collaborators for the comprehensive health examination. Third, we assessed nutrition data from BDHQ and did not examine actual diets. Finally, we did not obtain information on other variables, such as the use of supplementations, hours of sunlight, season and history of drug use.

## 5. Conclusions

We performed a cross-sectional study with Japanese representatives to examine the association between dietary calcium intake and hypertension. High intake of calcium was inversely associated with hypertension in subjects with serum concentrations of 25(OH)D higher than 20 ng/mL. Furthermore, a correlation was observed between dietary vitamin D intake and serum concentrations of 25(OH)D. The regular dietary consumption of calcium may contribute to the prevention and treatment of hypertension in subjects with a non-vitamin D deficiency and dietary vitamin D intake represents one of the valid approaches for avoiding this deficiency. 

## Figures and Tables

**Figure 1 nutrients-11-00911-f001:**
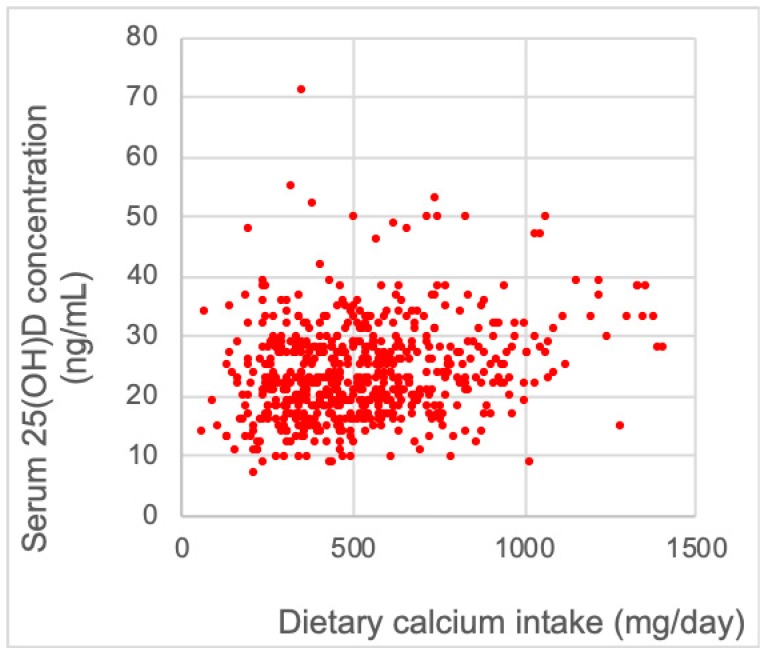
Association between dietary vitamin D intake and serum 25(OH)D concentration. The spread of the association between dietary calcium intake and serum 25(OH)D levels (*n* = 619). Pearson’s correlation coefficient was 0.285 (*p* < 0.001).

**Table 1 nutrients-11-00911-t001:** Characteristics of subjects in different blood pressure (BP) groups.

	All Participants	Hypertension	Normal BP	*p* Value
No. of subjects	619	343	276	
Men, *n* (%)	292 (47)	191 (56)	101 (37)	**<0.001**
Age	60.9 (11.5)	64.4 (10.7)	56.7 (11.0)	**<0.001**
Smoking status, *n* (%)				**0.034**
Non-smoker	326 (53)	165 (48)	161 (58)	
Ex-smoker	167 (27)	104 (30)	63 (23)	
Current	126 (20)	74 (22)	52 (19)	
Exercise habit, *n* (%)				0.393
yes	337 (54)	192 (56)	145 (53)	
no	282 (46)	151 (44)	131 (47)	
Height (cm)	160 (9.66)	160 (9.84)	160 (9.44)	0.920
Weight (kg)	59.8 (11.7)	61.5 (11.4)	57.8 (11.7)	**<0.001**
Waist circumference (cm)	83.7 (9.16)	85.5 (8.90)	81.5 (8.98)	**<0.001**
BMI (kg/m^2^)	23.1 (3.22)	23.8 (3.12)	22.4 (3.17)	**<0.001**
SBP (mmHg)	139 (20.2)	151 (18.2)	124 (9.13)	**<0.001**
DBP (mmHg)	80.7 (11.8)	85.7 (12.2)	74.5 (7.51)	**<0.001**
eGFR (mL/min/1.73m^2^)	72.6 (14.4)	70.7 (15.3)	75.0 (12.8)	**<0.001**
Serum 25(OH)D (ng/mL)	23.8 (8.04)	24.7 (7.95)	22.6 (8.01)	**0.001**
Energy and nutrients (/day)				
Total energy (kcal)	1856 (614)	1885 (622)	1819 (603)	0.182
Nutrients (g/1000 kcal)				
Protein	38.0 (8.16)	37.9 (8.66)	38.2 (7.52)	0.660
Carbohydrate	134.1 (21.2)	134 (22.4)	134 (19.6)	0.806
Sodium	2.44 (0.541)	2.49 (0.544)	2.39 (0.533)	**0.026**
Potassium	1.38 (0.426)	1.39 (0.451)	1.38 (0.392)	0.889
Magnesium	0.140 (0.330)	0.141 (0.035)	0.140 (0.31)	0.576
Calcium (mg/1000 kcal)	292 (111)	292 (113)	292 (110)	0.991
Vitamin D (μg/1000 kcal)	8.41 (5.26)	8.72 (5.56)	8.02 (4.85)	0.091
Cholesterol	0.202 (0.723)	0.197 (0.750)	0.207 (0.686)	0.086
Total dietary fibre	6.51 (2.18)	6.57 (2.29)	6.44 (2.04)	0.447
Alcohol	6.99 (10.5)	8.59 (11.4)	4.99 (9.08)	**<0.001**
SFA	7.26 (2.17)	6.87 (2.07)	7.74 (2.20)	**<0.001**
MUFA	9.77 (2.66)	9.35 (2.70)	10.3 (2.51)	**<0.001**
PUFA	6.82 (1.75)	6.62 (1.77)	7.07 (1.68)	**0.002**
*n*-3 fatty acids	1.48 (0.504)	1.48 (0.508)	1.49 (0.500)	0.732
*n*-6 fatty acids	5.31 (1.41)	5.12 (1.44)	5.55 (1.34)	**<0.001**

*p* values were from the Student’s *t*-test for continuous variables and from the chi-square test for categorical variables. *p* values < 0.05 are in bold. Continuous variables are presented as the mean (SD). Nutrient data were adjusted for energy using the density method as a percentage of daily energy intake. Hypertension was defined as the use of antihypertensive medication or a BP of 140/90 mmHg or higher. Abbreviations: BP, blood pressure; BMI, body mass index; SBP, systolic blood pressure; DBP, diastolic blood pressure; eGFR, estimated glomerular filtration rate; 25(OH)D, 25-hydroxyvitamin D; SAF, saturated fatty acids; MUFA, monounsaturated fatty acids; PUFA, polyunsaturated fatty acids.

**Table 2 nutrients-11-00911-t002:** Interaction between BP groups and serum 25(OH)D levels for calcium intake.

Nutrient Intake	25(OH)D Level	HTN	NBP	*p* for Interaction
		Mean (SD)	Mean (SD)	
Calcium (mg/1000 kcal)	Low	291 (118)	263 (91.8)	**0.015**
	Normal	293 (111)	311 (117)	

*p* values for the interaction from a two-way analysis of variance. *p* value <0.05 is in bold. Nutrient data were adjusted for energy using the density method as a percentage of daily energy intake. Hypertension was defined as the use of antihypertensive medication or a BP of 140/90 mmHg or higher. A low serum 25(OH)D level was defined as a 25(OH)D value of 20 ng/mL or lower. A total of 201 subjects had a low serum 25(OH)D level; 110 had normal BP and 91 had hypertension. There were 418 subjects with a normal serum 25(OH)D level; 166 had a normal BP and 252 had hypertension. Abbreviations: BP, blood pressure; HTN, hypertension; NBP, normal blood pressure; 25(OH)D, 25-hydroxyvitamin D.

**Table 3 nutrients-11-00911-t003:** Association between calcium intake and blood pressure according to serum 25(OH)D levels.

Subjects	Model	OR	95% CI	
			Lower	Upper
All subjects (*n* = 619)	Model 1	0.998	0.996	1.000
	Model 2	0.997	0.994	1.001
	Model 3	0.997	0.994	1.001
Normal 25(OH)D level (*n* = 418)	Model 1	**0.997**	**0.995**	**0.999**
	Model 2	**0.995**	**0.991**	**0.999**
	Model 3	**0.995**	**0.991**	**0.999**
Low 25(OH)D level (*n* = 201)	Model 1	1.002	0.998	1.006
	Model 2	1.003	0.996	1.011
	Model 3	1.003	0.995	1.010

Statistically significant estimates are in bold. Nutrient data were adjusted for energy using the density method as a percentage of daily energy intake. Model 1: adjusted for sex, age, BMI, eGFR, frequency of exercise, smoking status, consumption of alcohol and sodium intake. Model 2: adjusted for sex, age, BMI, eGFR, frequency of exercise, smoking status, sodium intake and the consumption of alcohol, proteins, carbohydrates, total dietary fibre, saturated fatty acids, monounsaturated fatty acids, protein, *n*-3 fatty acids and *n*-6 fatty acids. Model 3: adjusted for sex, age, BMI, eGFR, frequency of exercise, smoking status, serum 25(OH)D, the consumption of alcohol, protein, carbohydrates, total dietary fibre, saturated fatty acids, monounsaturated fatty acids protein, *n*-3 fatty acids and *n*-6 fatty acids and sodium intake. Abbreviations: OR, Odds ratio; CI, confidence interval; eGFR, estimated glomerular filtration rate; 25(OH)D, 25-hydroxyvitamin D.

**Table 4 nutrients-11-00911-t004:** Association between dietary vitamin D intake and serum 25(OH)D.

Model	β	95% CI	
		Lower	Upper
Model 1	**0.182**	**0.108**	**0.255**
Model 2	**0.179**	**0.105**	**0.252**
Model 3	**0.178**	**0.105**	**0.251**
Model 4	**0.184**	**0.112**	**0.257**

Statistically significant estimates are in bold. All data were transformed to z scores. Nutrient data were adjusted for energy using the density method as a percentage of daily energy intake. Model 1: adjusted for sex, age. Model 2: adjusted for sex, age, BMI. Model 3: adjusted for sex, age, BMI, exercise habit, alcohol intake, smoking status. Model 4: adjusted for sex, age, BMI, exercise habit, alcohol intake, smoking status, eGFR. Abbreviations: eGFR, estimated glomerular filtration rate; 25(OH)D, 25-hydroxyvitamin D.

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
