# Peer review of "Dietary Calcium Intake and Hypertension: Importance of Serum Concentrations of 25-Hydroxyvitamin D"

_nutrients, 2019, doi:10.3390/nu11040911_

Round 1

Reviewer 1 Report

I congratulate the authors for coming out with this subject and explore the role of calcium and vitamin D on blood pressure. The article is very interesting and contributes to the literature in this field of research. Overall, it has a fluid reading, however, some changes need to be addressed and a few concerns need to be explained.

General topics:

When specifically stating about measures of association, I suggest the authors to use “association” instead of “relation/relationship” (i.e., the association between the exposure and the outcome, or the association of the exposure with the outcome).

It is important to have present that the major contributor for the vitamin D status is vitamin D from cutaneous synthesis; at least 80% of this state is from the endogenous production.

The description of the objective of the study should be consistent among the several sections of the article (i.e., abstract, introduction, discussion, conclusion).

Title

I think that the semicolon should not be in the tittle.

Abstract

Taking into consideration the general comment, please review the objective.

I suggest change this phrase:

A high dietary calcium intake inversely correlated with hypertension in subjects with serum 25(OH)D levels of higher than 20 ng/mL (odds ratio, 0.995; 95% confidence interval. 0.991 to 0.999), while this relationship was not significant in those with serum 25(OH)D levels of 20 ng/mL or lower.

To:

Dietary calcium intake inversely correlated with hypertension in subjects with serum 25(OH)D levels higher than 20 ng/mL (OR: 0.995; 95% CI: 0.991, 0.999), but it was not significant in those with serum 25(OH)D levels of 20 ng/mL or lower.

Introduction

Line 36: such as a reduced sodium – remove the a

I suggest change this:

Increases in calcium intake have been shown to lower BP in hypertensive and even normotensive subjects. Furthermore, dietary calcium interventions

To:

Increases in calcium intake have been shown to lower BP in both normotensive and hypertensive subjects. Dietary calcium interventions

Line 43: that a higher calcium intake from food and supplements is associated with a lower risk of – remove the a

Line 45: please briefly state the mechanisms involved in the effect of calcium on BP

I suggest change this:

A close relationship exists between calcium intake and vitamin D; vitamin D intake improves calcium absorption and inhibits its excretion from the kidneys [8]. For example, osteoporosis, which is characterized by a low bone mass and the deterioration of bone tissue that increase the risk of bone fractures, is mainly associated with an inadequate calcium intake; however, an insufficient vitamin D intake also contributes to osteoporosis due to the associated reductions in calcium absorption [9].

To:

A close relationship exists between calcium and vitamin D. A sufficient vitamin D status promotes the intestinal calcium absorption and inhibits its excretion from the kidneys [8]. For example, osteoporosis, which is characterized by low bone mass and deterioration of bone tissue, is mainly associated with an inadequate calcium intake. However, an insufficient vitamin D status also contributes to osteoporosis due to the associated reductions in calcium absorption [9].

I suggest change this:

To elucidate the relationship between dietary calcium and hypertension, it is important to consider the effects of vitamin D; however, few studies have investigated this triangular

To:

To elucidate the relationship between dietary calcium and hypertension, it is important to consider the role of vitamin D. However, few studies have investigated this triangular

I suggest to remove this information from introduction; it can be presented in the discussion:

The joint association of dietary calcium intake and dietary vitamin D intake with the risk of hypertension was also investigated in this study, but was not significant.

I suggest to delete this: Since vitamin D is produced endogenously when ultraviolet rays from sunlight strike the skin and trigger vitamin D synthesis, apart from dietary vitamin D intake, serum concentrations of 25-hydroxyvitamin D [25(OH)D] are the best indicator of the vitamin D status [8].

And add this: Dietary vitamin D intake and vitamin D from cutaneous synthesis reflect on serum concentrations of 25(OH)D, which is the biomarker of the vitamin D status [8].

I suggest change this:

Therefore, the present cross-sectional study was performed in an attempt to elucidate the

To:

Therefore, the present cross-sectional study aimed to examine the

Methods

Study design and participants

Line 75: The target subjects of the SHIKA study was all – change to were

I suggest change this:

All subjects in the model districts (n = 2160) were delivered a self-administrated questionnaire by interviewers who were trained in their use. Interviewers explained the outline

To:

All subjects in the model districts (n = 2160) were delivered a self-administrated questionnaire by the researchers, who explained the outline

Serum 25(OH)D measurement

I suggest change this:

Serum concentrations of 25(OH)D were used as an indicator of the vitamin D status

To:

Serum concentrations of 25(OH)D were used as the biomarker of the vitamin D status

And, thus delete this: Serum concentrations of 25(OH)D are regarded as the most accurate indicator of the vitamin D status.

I suggest change this:

In the present study, a low serum 25(OH)D concentration was defined as a 25(OH)D value of 20 ng/mL (50 nmol/L) or lower; this cut-off point is generally considered to be inadequate for bone and overall heath in healthy subjects and the main definition of a vitamin D deficiency [8, 21].

To:

According to the Institute of Medicine, serum concentration of 25(OH)D ≥20ng/mL (50 nmol/L) is sufficient for maintaining bone health [8]. In the present study, low serum 25(OH)D concentration was defined as a 25(OH)D value of 20 ng/mL (50 nmol/L) or lower.

Statistical analysis

Line 140: chi-squared – change to chi-square

Line 148: relationship – change to association

Regarding the paragraph below, I have two concerns. The first is that there is no data on season, which is a potential confounder. The second is that I did not see these results in the manuscript. In contrast, in the results section it is presented the table 4 with data on the association between dietary vitamin D intake and serum 25(OH)D, which is not mentioned in the statistical analysis. The authors should clarify this.

Furthermore, to evaluate the relationship between serum concentrations of 25(OH)D and dietary calcium intake, we used a multiple linear regression analysis controlling for sex, age, BMI, eGFR, smoking status, and exercise habit.

I suggest change this:

In all analyses, the threshold for significance was P<0.05. All statistical analyses were performed

To:

Statistical significance was considered with an α critical value of 0.05. Statistical analyses were performed

Ethics statement

Informed consent was obtained from all participants – signed inform consent? Please add this information.

Results

Line 158: according to hypertension were shown – change to according to the BP groups are shown

Line 159: were 139 – change to were 139 mmHg

Line 162: and had a lower eGFR – remove the a

Table 1: I suggest the authors to put the statistical significant p-values in bold; and add to the caption: p-values <0.05 are in bold. Also, it would be more interesting to separate non- and ex-smokers.

I suggest change this:

groups according to 25(OH)D groups, (Table 2). The prevalence of a low serum 25(OH)D level (25(OH)D value of 20 ng/mL or lower) was 32% (n = 201). The results obtained revealed a significant

To:

groups according to 25(OH)D groups (Table 2). The prevalence of low serum 25(OH)D level was 32% (n = 201). The obtained results revealed a significant

Table 2: I suggest the authors to put the statistical significant p-value in bold; and add to the caption: p-value <0.05 is in bold.

Line 193: to evaluate the relationship between dietary calcium intake and hypertension. I suggest change to: to evaluate the association between dietary calcium intake and blood pressure groups

Line 196: results obtained were shown in Table 3 – change to are

Table 3: change relationship for association. Take off the bold and the line from All subjects Model 1 0.998. Remove the p-value column – the 95% CI is sufficient. I suggest the authors to put in bold the statistical significant OR and 95% CI and add to the caption: statistical significant estimates are in bold. In the caption, there are repeated information on the adjustment variables. For the different models the authors can delete P values from a multiple logistic regression analysis after adjustments for the following independent factors and just write Model x: adjusted for…

I suggest change levels to concentration: Figure 1 shows the spread of the relationship between dietary vitamin D and serum 25(OH)D levels. A correlation was observed between dietary vitamin D and serum 25(OH)D levels

Table 4: change relationship for association. Take off the bold and the line from Model 1 0.182. Remove the p-value column – the 95% CI is sufficient. I suggest the authors to put in bold the statistical significant β and 95% CI and add to the caption: statistical significant estimates are in bold. In the caption, there are repeated information on the adjustment variables. For the different models the authors can delete P values from a multiple linear regression analysis after adjustments for the following independent factors and just write Model x: adjusted for…The first phrase: β means a standardized regression coefficient can also be deleted. As stated in the comments to the statistical analysis, the associations presented in this table are not mentioned in the statistical analysis section. The authors should clarify this. All data were transformed to z scores – this information should also be stated in the statistical analysis, along with the justification for the transformation.

Discussion

Line 244: as an indicator – change to: as the biomarker

Line 245: that a high dietary – change to: that high dietary

Line 246: with a normal serum 25(OH)D level only – change to: with normal serum 25(OH)D level.

Lines 249/250: I think that phrase is a repetition of the phrase in lines 245/246, so it can be deleted.

I suggest change this:

Although the odds ratio of dietary calcium intake and hypertension in the present study was small and the reductions observed in BP by an increased calcium intake were also small in previous studies, even small reductions in BP and the risk of hypertension have important health implications for cardiovascular diseases.

To:

Although the odds ratio of dietary calcium intake and hypertension in the present study was small, in previous studies the reductions observed in BP by an increased calcium intake were also small, and small reductions in BP and in the risk of hypertension have important health implications for cardiovascular diseases.

Line 289: was only observed in subjects with a normal serum – change to: was only observed in the subjects with normal serum

Line 296: A vitamin D deficiency and insufficiency are common in many populations worldwide – change to: Vitamin D insufficiency is common in many populations worldwide.

Line 299: The prevalence of a vitamin D – change to: The prevalence of the risk of vitamin D

Line 306: I suggest doing paragraph.

Line 314: This result supports the importance and effectiveness of an aggressive dietary vitamin D intake in – change to: This result supports the importance of a high dietary vitamin D intake in

Line 315: for preventing a vitamin D deficiency – change to: for preventing vitamin D deficiency.

Line 321: and a history – change to: and history

Conclusion

I suggest change this phrase:

A high intake of calcium was inversely associated with hypertension in subjects with a serum concentration of 25(OH)D of higher than 20 ng/mL only.

To:

High intake of calcium was inversely associated with hypertension in subjects with serum concentration of 25(OH)D higher than 20 ng/mL.

References

The authors should uniform the references. The presentation of journals’ name should be consistent among references – also, the abbreviated names should be the ones defined by each journal -, as well as the usage of upper and lowercase in the articles’ title.

Suggested readings (facultative):

I share with the authors the following articles, than can be useful for this or future works.

Vimaleswaran, K. S., et al. (2014). "Association of vitamin D status with arterial blood pressure and hypertension risk: a mendelian randomisation study." Lancet Diabetes Endocrinol 2(9): 719-729.

Beveridge, L. A., et al. (2015). "Effect of vitamin D supplementation on blood pressure: a systematic review and meta-analysis incorporating individual patient data." JAMA Intern Med 175(5): 745-754.

Pilz, S., et al. (2016). "Vitamin D and cardiovascular disease prevention." Nat Rev Cardiol 13(7): 404-417.

Collaboration, N. C. D. R. F. (2017). "Worldwide trends in blood pressure from 1975 to 2015: a pooled analysis of 1479 population-based measurement studies with 19.1 million participants." Lancet 389(10064): 37-55.

Author Response

We greatly appreciate your review of our manuscript and the provision of helpful suggestions. The attached Word file is our responses to the reviewers’ comments, with a description of the changes made to the manuscript. In the revised manuscript, red text indicates the portions revised according to the comments of Reviewer 1.

  Thank you in advance for considering our revised manuscript for publication in Nutrients.

Reviewer 2 Report

Thank you for your contribution to our journal. However, there are some issues to consider.

619 subjects are not enough to represent the main issue in your paper to elucidate the relation between dietary calcium and HTN in normal 25OHD level.

Small difference of dietary calicum intake, 293mg/1000kcal vs. 311mg/1000kcal in HTN vs. NBP in normal 25OHD groups may be hard to say the real difference, even though the table 3 adjusted many factors.

Author Response

We greatly appreciate your review of our manuscript and the provision of helpful suggestions. The attached Word file is our responses to the reviewers’ comments, with a description of the changes made to the manuscript. In the revised manuscript, blue text indicates the portions revised according to the comments of Reviewer 2.

  Thank you in advance for considering our revised manuscript for publication in Nutrients.
